# Gut Microbiome Remains Static in Functional Abdominal Pain Disorders Patients Compared to Controls: Potential for Diagnostic Tools

**DOI:** 10.3390/biotech11040050

**Published:** 2022-10-27

**Authors:** Bassam Abomoelak, Miguel Saps, Sailendharan Sudakaran, Chirajyoti Deb, Devendra Mehta

**Affiliations:** 1Arnold Palmer Pediatric Gastroenterology Clinic, Orlando Health, Orlando, FL 32806, USA; 2Pediatric Gastroenterology, Hepatology and Nutrition Division, University of Miami Miller School of Medicine, Miami, FL 33136, USA; 3Wisconsin Institute for Discovery, University of Wisconsin, Madison, WI 53715, USA

**Keywords:** gut microbiome, functional abdominal pain disorders, bacterial dysbiosis

## Abstract

Background: Functional Abdominal Pain disorders (FAPDs) are a group of heterogeneous gastrointestinal disorders with unclear pathophysiology. In children, FAPDs are more common in the winter months than summer months. The possible influence of school stressors has been proposed. Previously, our group showed differences in bacterial relative abundances and alpha diversity in the gut microbiome and its relationship with stressors in a cross-sectional evaluation of children suffering from FAPDs compared to a healthy control group. We present longitudinal data to assess whether the gut microbiome changes over school terms in the control and FAPDs groups. Methods: The longitudinal study included children with FAPDs (*n* = 28) and healthy controls (*n* = 54). Gastrointestinal symptoms, as well as stool microbiome, were assessed in both groups. Stool samples were serially collected from all participants during both the school term and summer vacation. The stool samples were subjected to total genomic extraction, 16S rRNA amplicon sequencing, and bioinformatics analysis. The gut microbiome was compared at school and during vacation. Other metrics, alpha diversity, and beta diversity, were also compared between the two school terms in every group. Results: In the healthy group, there were differences in microbiome composition between school terms and summer vacation. Conversely, we found no differences in the FAPDs group between the two terms. The healthy control group revealed differences (*p*-value < 0.05) in 55 bacterial species between the school term and vacation. Several of the differentially abundant identified bacteria were involved in short-chain fatty acids production (SCFAs), inflammation reduction, and gut homeostasis. Alpha diversity metrics, such as the Shannon index, were different in the control group and remained unchanged in the FAPDs group. Conclusion: Although preliminary, our findings suggest that the gut microbiome is static in FAPDs. This compares with a more dynamic healthy gut microbiome. Further studies are warranted to corroborate this and understand the interplay between stress, symptoms, and a less diverse and static microbiome. Future studies will also account for different variables such as diet and other patient demographic criteria that were missing in the current study.

## 1. Introduction

Functional Abdominal Pain disorders (FAPDs), a group of heterogeneous gastrointestinal disorders, are common among schoolchildren. Children with FAPDs frequently report comorbid extraintestinal symptoms and limitations in social and physical activities [1]. FAPDs, as a group, include four distinct diagnoses: functional dyspepsia, irritable bowel syndrome (IBS), functional abdominal pain—NOS (not otherwise specified), and abdominal migraine [2,3]. To date, the pathophysiology of FAPDs remains largely unknown. The most accepted model to explain FAPDs proposes that those result from the interaction of multiple factors, including stressors, diet, and microbiome. The importance of gut microbiota is increasingly recognized and is now considered central to the pathophysiology of FAPDS. Few studies have analyzed the microbiota of children with FAPDs [4,5]. Our group has previously studied the microbiome of children with FAPDs and healthy controls. We found that the gut microbiome of children with FAPDs differed from the healthy gut microbiome [6]. The prospective cohort study found differences in children suffering from FAPDs compared to the healthy gut microbiome in terms of bacterial relative abundances at the phylum and genus levels [6]. The three main phyla (Bacteriodetes, Firmicutes, and Verrucomicrobiota) were significantly different between the control and FAPDs groups [6]. At the genus level, several gut bacteria also showed differential abundances between the two groups. In addition, other metrics such as α-diversity and β-diversity differed the FAPDs group from the healthy gut microbiome [6]. Unlike in adults, FAPDs in children follow a seasonal pattern [7]. A retrospective cohort study that analyzed data from six tertiary care institutions in the US found that the rates of abdominal pain consultations in children were consistently higher in winter months [7,8]. The factors associated with this winter predominance have not been identified. Some authors explain the seasonality by changes in weather and how it influences the child’s physical activity of children, while others attribute it to the differential effect of stressors during the school terms [1,7,8,9]. Stress has been implicated in altering the fecal microbiome in children in general and adults with and without IBS [10,11]. Stress levels vary between children during school and vacations [12]. Diet, sleep, and exercise are also influenced by school terms and vacations. All these factors were found to influence the child’s symptoms as well as the gut microbiome [13,14,15,16]. No studies have sequentially investigated the microbiome of children with FAPDs during school and vacation. Changes in the microbiome may be a contributing factor to the differences in symptoms found in children with FAPDs between the school term and vacation. A better understanding of the gut microbiome across time may help advance our understanding of the pathophysiological factors involved in the development of symptoms in children with FAPDs and could potentially help identify biomarkers to guide diagnosis and treatment. In the current study, we aimed to investigate the alterations in the gut microbiome of children with and without FAPDs over time. We compared the gut microbiome in FAPDs and healthy control subjects between school and vacation terms. 

## 2. Materials and Methods

### 2.1. Recruitment and Study Design

This was a longitudinal study. Children with and without FAPDs (healthy controls) who met the inclusion/exclusion criteria participated in the study. Children with FAPDs (cases) were recruited from a large pediatric gastroenterology clinic in Central Florida. The healthy controls were recruited from five local pediatric practices in the greater Orlando area. The control group included healthy children seen for routine exams or well-check appointments who met the inclusion/exclusion criteria. The participants consented via child and/or parental consent based on age. Figure 1 depicts the flowchart of the recruitment and group assignment. Inclusion criteria were children diagnosed with FAPDs according to the Rome IV criteria [17]. The exclusion criteria included acute illnesses, such as appendicitis, gastroenteritis, and chronic diseases, such as cancer, inflammatory bowel disease (IBD), coeliac disease, and stomach ulcer. In addition, children with current febrile illness or who were actively taking antibiotics or had completed a course of antibiotics two-week before enrollment were also excluded from the study. Research forms were managed using REDCap electronic data capture tools hosted at Florida State University [18]. The project was approved by FSU Institutional Review Board (IRB) ethics committee (One Florida, Study ID: IRB 201701009).

### 2.2. Stool Samples Collection and Sequencing

The patients and caregivers were instructed to collect three stool samples at home and send them via US mail. The hemoccult ICT collection kit was used to collect all of the stool samples (Beckman Coulter, Brea, CA, USA). The kit included a toilet hat, application stick, collection card, collection pouch, specimen biohazard bag, pre-addressed mailing envelope, and detailed paper instructions about the collection procedures. The procedure has been validated as a cost-effective method of stool collection with no significant microbiome differences from other stool collection methods [19]. The stool samples were frozen at −20 °C until further use. The DNA was sequenced using Illumina MiSeq, and the bioinformatics analysis was performed as previously described [6]. Briefly, the microbiome analysis was performed by the UW biotechnology center using Quantitative Insights Into Microbial Ecology (QIIME2) version 2, as previously described [6,20]. Illumina sequencing reads were denoised and quality filtered using the denoising program DADA2 [6]. This step trimmed low-quality bases, filtered out noisy sequences, corrected errors in marginal sequences, removed chimeric sequences and singletons, joined denoised paired-end reads, and then dereplicated those sequences. The low-frequency reads (<0.01%) were filtered from the Biome-formatted table. Alpha rarefaction curves using Shannon, Simpson, and Observed-species were calculated for all samples with a rarefaction upper limit of median depth/sample count, and the alpha diversity between the different treatments was compared using Wilcoxon signed-rank test. Beta diversity was calculated, and ordination plots were generated using Bray–Curtis and Jaccard (Non-Phylogenetic), weighted and unweighted Unifrac (Phylogenetic) on ASV data leveled according to the lowest sample depth.

The follow-up comparative analysis such as Alpha and beta diversity metrics were assessed using R with the packages such as phyloseq, tidyverse, ggplot2 [21,22,23,24].

### 2.3. Statistical Analyses

Multivariate analysis by linear models (MaASLin) was used to assess the differences in the relative abundance of the bacterial phyla between the groups (FAPDs vs. controls) and for the academic year (school vs. vacation). GraphPad Prism V.8 (Version 8.3.1., San Diego, CA, USA) was also used for the analyses of the bacterial relative abundances between groups; two-tailed, and a *p* < 0.05 was used for statistical significance. (MaASLin) statistical pipelines from Huttenhower lab Galaxy (http://huttenhower.sph.harvard.edu/galaxy) (Access date 15 June 2022) were used in the analysis of the data. 

## 3. Results

### 3.1. Participants’ Demographics

Forty-five children with FAPDs and 86 healthy were enrolled in the study (Figure 1). Twenty-eight children with FAPDs and 54 healthy controls completed the questionnaires and provided all necessary stool samples for the study (age 7 to 16 years, mean age 11 ± 2.58 years). Fifty-two percent of the participants were females (43/82), and 61% were non-Hispanic. The two groups did not reveal age or race differences (*p*-value > 0.05). Sixty-five percent of the enrolled participants did not take antibiotics in the year before the study, and none of them received antibiotics two weeks prior. Briefly, both groups were similar in age (*p* = 0.287), sex (*p* = 0.64), and race/ethnicity (*p* = 0.400). Table 1 depicts the classification of 23 FAPDs patients, which was performed by an experienced gastroenterologist from Orlando Health according to Rome IV classification. Noteworthy, some FAPDs patients showed more than one diagnosis. The healthy control group provided a total of 143 stool samples (school term = 123, vacation = 20). The FAPDs group provided a total of 70 stool samples in both school terms (school = 20, vacation = 50). For longitudinal assessment, a subset of samples with matching pairs between the school term and vacation term was analyzed, and this subset included 11 children with FAPDs and 19 controls.

### 3.2. Bacterial Relative Abundances within Groups during School Terms and Vacation

The analysis of the FAPDs and healthy control groups were conducted separately. We used MaAsLin software to compare the two groups in the school and vacation terms using all stools as independent samples. The comparison of FAPDs gut microbiome in school and vacation revealed that *Enterobacter* was the only identified bacteria to be statistically and significantly different (*p*-value < 0.001) (data not shown). In contrast, the healthy group gut microbiome showed that at least 55 bacterial species revealed statistically significant differences (*p*-value < 0.05) (Table 2). 

Among the 55 identified bacterial species in the control group, only nine were decreased in vacation in comparison to the school term. These nine bacterial species included *Holdemania*, *Bacteroides*, *Collinsella*, *Lachnospiraceae*, *Fusicatenibacter*, *Enterococcus*, and *Alistipes* (Table 2). We used Mann–Whitney U to confirm the differences for some bacterial species. Of the two most abundant species, *Bacteroides* (*p*-value < 0.001), but not *Faecalibacterium*, was lower in the vacation term compared to the school. In addition, *Alistipes* levels (*p*-value < 0.05) were also lower during vacation, while *Prevotella* levels were higher (*p*-value < 0.001). In addition, genera such as *Rothia*, *Lactobacillus*, and *Streptococcus* were higher in vacation terms compared to the school term (*p*-value < 0.001) in controls but not in the FAPDS group (Figure 2). 

In subsets of FAPDs and the controls with paired samples available, we analyzed with one sample (the first one if more than one was available) in school and one on vacation, excluding repetitive samples of the same subject in each school term. This selection reduced the sample size to 19 paired samples for the control group and 11 paired samples for the FAPDs group. Using paired testing, we found that the FAPDs group did not reveal differences in any bacterial taxa, while the healthy control group had approximately 52 bacterial taxa that were significantly different. The two most abundant bacterial species in the gut, *Bacteroides* and *Faecalibacterium* showed statistically significant differences in the healthy control group (Mann–Whitney U, *p*-value < 0.05), while no significant changes were observed in the FAPDs group. *Lactobacillus* showed a similar pattern as well, while the *Bifidobacterium* and *Ruminococcus* levels were unaltered in both groups. These findings suggest a static nature of the microbiome in the FAPDs compared to a dynamic, healthy control microbiome (Figure 3).

### 3.3. Alpha and Beta Diversity Comparison in FAPDs and Control Groups

We measured and compared α-diversity (Shannon diversity) between the school terms in the FAPDs and control groups. Shannon diversity did not reveal any statistically significant differences in the FAPDs group between the two school terms (*p*-value > 0.05), while a difference was observed in the control group (*p*-value < 0.05) (Figure 4A,B). Additionally, other α-diversity metrics such as Chao1, Simpson, and Observed_species showed consistent and similar results to the Shannon index (data not shown). Beta diversity was compared between the school term and vacation in the control and the FAPDs paired samples. The control group showed a clear clustering when both school terms were analyzed by principal coordinates analysis (PCA), while the FAPDs comparison did not reveal any clustering (Figure 5). PERMANOVA analysis showed a statistically significant difference only in the control group (*p*-value < 0.05). PCA of Bray–Curtis distances discriminated by school term in the control group with statistical significance (*p*-value = 0.01).

## 4. Discussion

Recent studies have suggested a possible role of gut microbiota in the pathophysiology of multiple gastrointestinal (GI) diseases such as IBS, inflammatory bowel disease (IBD) [25,26,27,28], obesity, and Type 1 and 2 diabetes [29]. Several studies have linked bacterial genera abundance to GI diseases. *Faecalibacterium* and *Dorea* showed altered abundances in the gut of IBS patients [30]. Similarly, in IBD patients, *Bacteroides* may be involved in the development of the disease [27]. Microbiome alterations not only cause local changes but also may result in central nervous system effects, which may explain some of the extraintestinal symptoms reported by children and adults [31]. Our group revealed that the FAPDs and healthy gut microbiomes are different in bacterial relative abundances and other metrics [6]. The role of the gut microbiome in the onset of the FAPDs symptoms remains largely unknown, but the link between the gut and the brain became a potential cause for some idiopathic diseases, such as IBS [32,33]. We previously confirmed that the FAPDs stress severity scores showed a correlation with some bacterial genera in a cross-sectional analysis, but the full spectrum of bacterial roles in the symptoms remains to be elucidated [6]. In the current study, we compared the longitudinal changes between patients who met Rome IV criteria for FAPDs and healthy controls. The healthy gut microbiome showed alterations between the school and vacation terms, while the FAPDs gut microbiome remain largely unchanged. Alterations in the alpha and beta diversity were also observed in the control group, while no significant changes were noticed in the FAPDs group. We had previously shown less diversity in the FAPDs group. This finding suggests that the reduced diversity is also static and indeed supports the concept that lower diversity reduces the capacity to adapt, as opposed to controls.

Studies in adults with irritable bowel syndrome [10] showed stress-related changes in firmicutes and Bacteroidetes, as well as *Ruminococcus*, which were previously implicated in adult IBS.

Indeed, the human gut is likely to be affected by environmental factors, including xenobiotics, stress, diet, and lifestyle throughout the individual’s lifespan, and is an important key player in IBS pathogenesis [34]. Generally, stress from schoolwork, for example, may lead to an increase in symptoms in a gut with a static, suboptimal, and less diverse microbiome than in a gut with a normal microbiome [35]. Overall, microbiome changes implicating mood in IBS have been shown [26]. In the healthy gut, approximately 18% of bacterial species showed alterations over time between school term and vacation terms. In adult IBS patients, the data showed that the microbial alpha diversity and beta diversity did not exhibit significant changes when compared between IBS patients with and without higher psychological distress [10]. This finding supports the concept that greater diversity allows a more robust, dynamic, and likely homeostatic response in controls than in those with FAPDs. Noteworthy, the stress and symptoms experienced by children diagnosed with FAPDs could be a direct effect of the lack of diversity and plasticity of the gut microbiome across time.

*Enterobacter* was the only genera that showed a significant increase during a vacation in both control and FAPDs fecal samples. Interestingly, in the control group, some bacteria species involved in short-chain fatty acids (SCFAs) were among the 55 identified bacterial species that changed during a vacation in the control group. *Lactobacillus* and *Prevotella* were among the acetate-producing bacteria, while others such as *Bifidobacteria*, *Akkermansia*, and *Ruminococcus* [36] did not show changes during vacation [37]. Acetate-producing bacteria are involved in the pH regulation of the gut, appetite control, and protection against pathogens. The propionate-producing bacteria included members of the phyla Bacteriodetes and Firmicutes, and more specifically, *Bacteroides* and *Veillonella* [38]. *Bacteroides* levels were significantly higher during school terms. When we further studied the smaller groups with paired samples, only the control samples showed changes in the two most abundant species, *Bacteroides* (increase) and *Faecalibacterium* (decrease), during the school term (Figure 3A). Interestingly *Veillonella*, also implicated in IBS, showed a small but significant decrease only in controls during the school term. Propionate-producing bacteria are involved in inflammation reduction, fighting cancer, and appetite regulation. *Veillonella* and *Akkermansia* (*Verrucomicrobia*) both produce propionic and acetic acids. In a cross-sectional study, higher levels of organic acid-producing flora, especially acetic and propionic acid, were found to be associated with IBS, including *Veillonella* and *Lactobacillus* [39]. The relationship between the gut microbiome and FAPDs, however, is likely to be multifactorial rather than dependent solely on SCFAs production [40,41]. Other potential factors may include other metabolites, proteases, virulence factors, and other host–microbe interactions [42,43,44]. The study has its own limitations, such as the small sample size and the lack of data about diet. Although the age and the gender data of the participants were collected, the authors admitted that some metadata such as BMI, weight, and height were missing in the current study. These important factors will be addressed in a future larger study. The clear distinction between the two cohorts needs to be confirmed in a larger sample size at the bacterial and metabolite levels. The potential to use this finding as a leading tool in diagnostics will largely depend on further exploratory studies. If confirmed, we can anticipate randomized clinical trials to better target FAPDs in childhood before evolving into IBS in adulthood.

## 5. Conclusions and Future Directions

The FAPDs gut microbiome is static compared to the healthy age-matched group. This finding needs to be confirmed in a multicenter and a larger sample size study. Noteworthy, the selected population represents an understudied segment of the population, and 25% of the children with FAPDs evolve into IBS in adulthood. The static nature of the gut microbiome of FAPDs patients can offer clues about the symptoms, such as the stress associated with the symptoms. Based on these findings, we can explore options to measure the bacterial differences or metabolites in the stools of FAPDs patients. We just launched a larger study in our hospital to explore the variations in the gut microbiome and stool metabolites levels. If the gut microbiome showed the same pattern in lack of diversification at the level of bacterial abundances, diversity, and fecal metabolites, we could devise targeted measurements of certain bacterial species, such as *Faecalibacterium* and *Bacteroides*. A potential and accurate tool such as Droplet Digital PCR (ddPCR) can be used to estimate the absolute load of such bacterial species. Additional diagnostics can also be used at the fecal metabolites, especially bacterial metabolites, to confirm the concomitant differences at the metabolomic levels as well. As a futuristic approach for diagnosis and treatment, we can target the gut microbiome with a research-based probiotics cocktail tailored to every FAPDs patient. We can use the differential bacterial abundances to diagnose and monitor the disease progression in children.

## Figures and Tables

**Figure 1 biotech-11-00050-f001:**
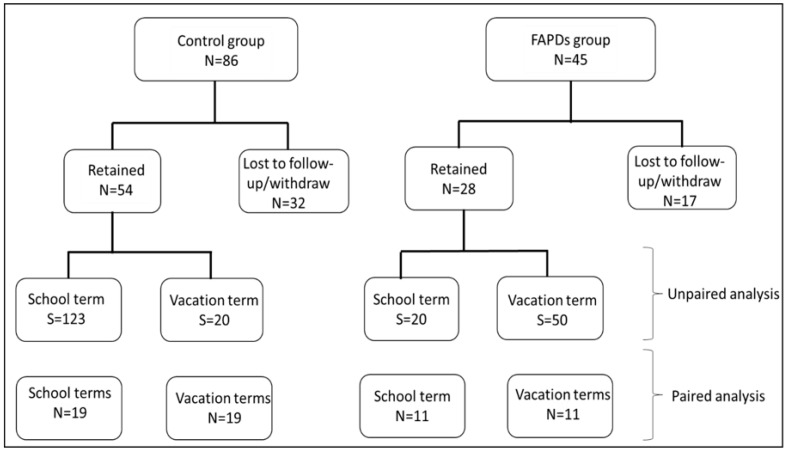
The flowchart of the participants in the study. The chart illustrates the total participants per group, loss of follow-up or withdrawal from the study, and retained patients. The analysis was separately performed on the unpaired and paired samples in every group. N denotes number of participants, while S denotes number of stool samples.

**Figure 2 biotech-11-00050-f002:**
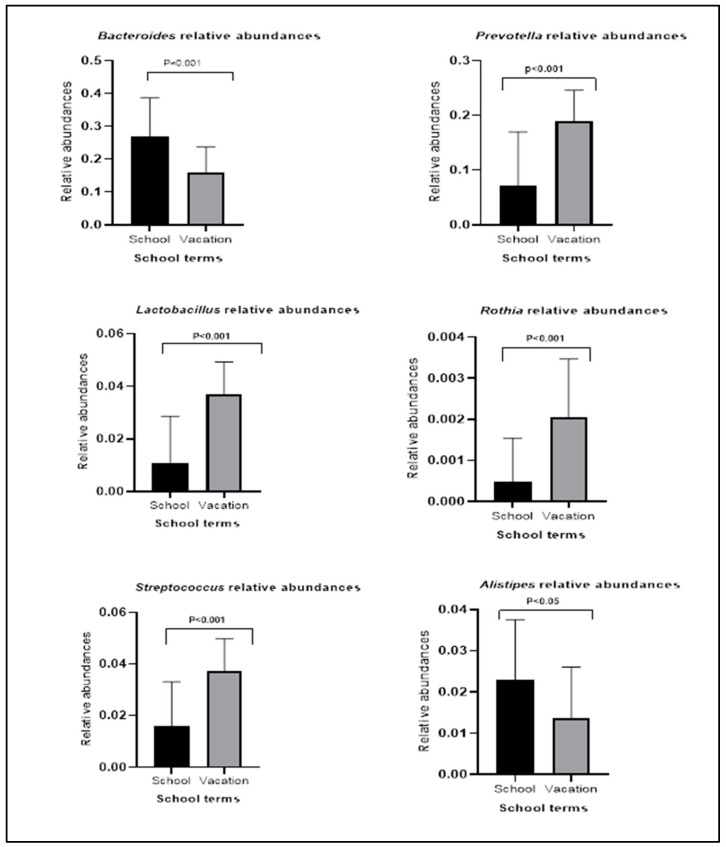
The relative abundance differences of different bacterial species in the control in school and vacation terms (*p* < 0.05 was considered significant). Mann–Whitney U test was used to compare the relative abundances in the control group (*p* < 0.05 was considered significant).

**Figure 3 biotech-11-00050-f003:**
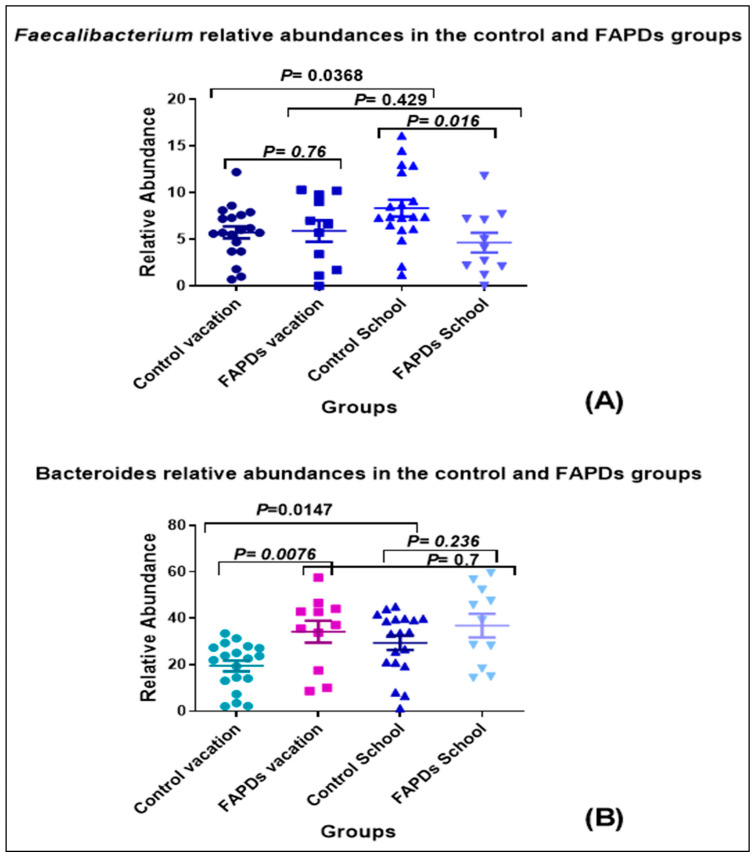
The relative abundance differences of different bacterial species in the control and FAPDs during the school and vacation terms (*p* < 0.05 was considered significant). Mann–Whitney U test was used to compare the relative abundances between control and FAPDs groups (*p* < 0.05 was considered significant). (**A**) denotes *Faecalibacterium* relative abundance, (**B**) denotes *Bacteroides* relative abundance, (**C**) denotes *Lactobacillus* relative abundance, while (**D**,**E**) denote *Bifidobacterium* and *Ruminococcus* relative abundances, respectively.

**Figure 4 biotech-11-00050-f004:**
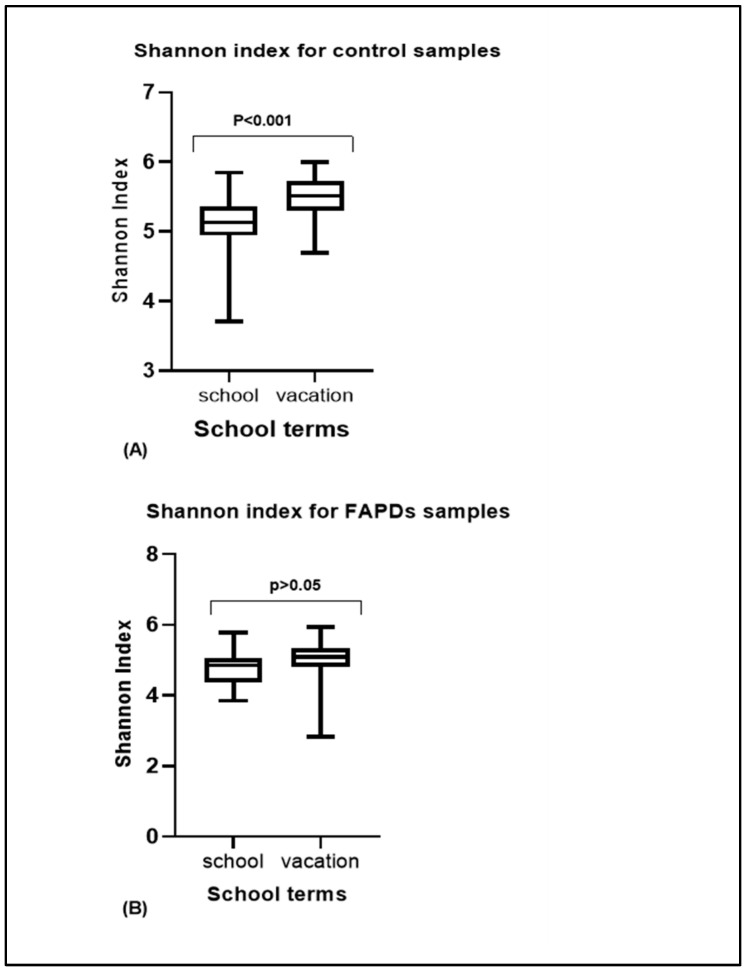
Alpha diversity (Shannon index) comparison between the control (**A**) and FAPDs (**B**) in the two school terms. *p* < 0.05 was considered significant.

**Figure 5 biotech-11-00050-f005:**
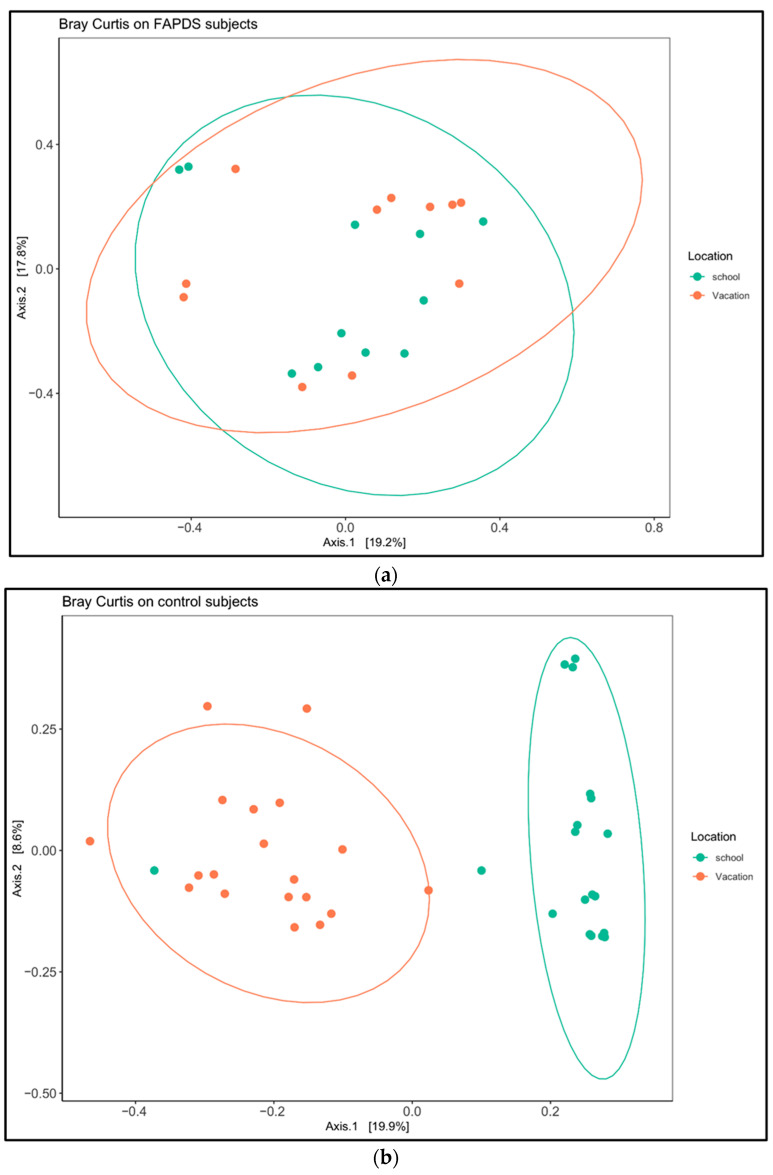
Beta diversity comparison in 11 paired FAPDs (**a**) patients and 19 paired control groups (**b**) at two different school terms. Bray–Curtis measurements showed statistical differences in the control group, while no statistical differences were observed in the FAPDs group.

**Table 1 biotech-11-00050-t001:** The classification of children diagnosed with FAPDs according to Rome IV criteria.

Rome IV Classification	Number of Patients (N = 23)
Irritable Bowel Syndrome	15
Functional Dyspepsia	13
Abdominal Migraines	7
Functional Constipation	5
Adolescent Rumination Syndrome	2
Functional Abdominal Pain—nos	1
Functional Nausea	1
Functional Vomiting	1
Cyclic Vomiting Syndrome	1
Non-retentive Fecal Incontinence	0
Aerophagia	0

**Table 2 biotech-11-00050-t002:** The list of the 55 differentially abundant bacteria in the control group during the two school terms. A *p*-value < 0.05 was considered significant. N is the number of samples, while N. not 0 implies the samples with relative abundances more than 0. Q value (False Discovery Rate, FDR) of <0.05 was considered significant. Coefficient denotes ratio between the two categories.

Feature	Coefficient	N	N Not 0	*p* Value	Q Value
*Bulleidia*	0.020327637	142	38	3.42E-11	8.05E-09
*Lactobacillus*	0.132172676	142	66	1.19E-09	1.35E-07
*Granulicatella*	0.048838228	142	59	2.15E-09	1.35E-07
*Fusobacterium*	0.050860561	142	60	2.29E-09	1.35E-07
*Alloprevotella*	0.051864263	142	52	4.50E-09	2.12E-07
*Neisseria*	0.069465138	142	58	5.68E-09	2.23E-07
*Atopobium*	0.051192828	142	53	1.89E-08	5.39E-07
*Solobacterium*	0.030782911	142	52	1.94E-08	5.39E-07
*Porphyromonas*	0.047079679	142	62	2.23E-08	5.39E-07
*Klebsiella*	0.071987738	142	59	2.29E-08	5.39E-07
*Moraxella*	0.033278381	142	51	2.59E-08	5.53E-07
*Oribacterium*	0.042789854	142	46	3.34E-08	6.40E-07
*Peptostreptococcus*	0.053773359	142	53	3.54E-08	6.40E-07
*Rothia*	0.029322402	142	37	3.93E-08	6.60E-07
*Enterobacteriaceae;__*	0.066945366	142	100	4.47E-08	7.01E-07
*Stenotrophomonas*	0.049565018	142	52	5.54E-08	8.14E-07
*Veillonella*	0.088241544	142	90	6.25E-08	8.64E-07
*Peptoniphilus*	0.047833579	142	53	1.03E-07	1.34E-06
*Prevotella*	0.265054965	142	90	2.68E-07	3.31E-06
*Parvimonas*	0.029579179	142	56	3.53E-07	4.14E-06
*Alloscardovia*	0.048653236	142	52	5.22E-07	5.85E-06
*Streptococcus*	0.083962421	142	133	6.72E-07	7.18E-06
*Megasphaera*	0.04917445	142	61	9.65E-07	9.86E-06
*Treponema*	0.009954054	142	23	1.35E-06	1.32E-05
*Gemella*	0.042810564	142	52	1.53E-06	1.43E-05
*Muribaculaceae*	0.014753749	142	33	2.51E-06	2.27E-05
*Leptotrichia*	0.015595742	142	36	2.61E-06	2.27E-05
Unassigned	0.019034108	142	41	3.07E-06	2.58E-05
*Candidatus_Saccharimonas*	0.0167842	142	27	4.47E-06	3.62E-05
*Aggregatibacter*	0.021560817	142	28	4.62E-06	3.62E-05
*Saccharimonadaceae;g__TM7x*	0.02284891	142	64	1.06E-05	8.01E-05
*Actinomyces*	0.033553175	142	48	1.21E-05	8.87E-05
*Stomatobaculum*	0.021394494	142	41	1.50E-05	0.000106562
*g__Lachnoanaerobaculum*	0.00686581	142	7	1.55E-05	0.000107388
*g__Mitochondria*	0.003454846	142	6	4.88E-05	0.000327673
*Campylobacter*	0.018845676	142	55	5.96E-05	0.000388877
*Bergeyella*	0.004791259	142	12	6.85E-05	0.000434922
*Bacilli;__;__;__*	0.025746385	142	43	7.79E-05	0.000481685
*Haemophilus*	0.036289019	142	84	9.31E-05	0.000561025
*Holdemania*	−0.017748516	142	87	0.000133085	0.000781876
*Selenomonas*	0.006838253	142	13	0.00022944	0.001315086
*Bacteroides*	−0.133075069	142	142	0.000238368	0.001332788
*Capnocytophaga*	0.010475722	142	26	0.000243872	0.001332788
*Collinsella*	−0.036662689	142	82	0.000279701	0.00149386
*[Eubacterium]_nodatum_group*	0.016523165	142	47	0.000393628	0.00205561
*Lachnospiraceae;__*	−0.034771487	142	137	0.002178163	0.011127572
*Absconditabacteriales_(SR1)*	0.005121259	142	19	0.002877646	0.014388232
*Pseudomonas*	0.001975578	142	3	0.005612792	0.027479295
*Fusicatenibacter*	−0.028887826	142	134	0.006120786	0.029354791
*Incertae_Sedis*	−0.024059973	142	139	0.006365965	0.029920034
*Enterococcus*	−0.016408838	142	38	0.007243531	0.033377053
*Alistipes*	−0.036479155	142	135	0.008722811	0.039420396
*Ochrobactrum*	0.004726509	142	21	0.009257382	0.041046884
*_Family_XIII_AD3011_group*	−0.017485629	142	87	0.010148283	0.044163825
*Anaeroglobus*	0.005091565	142	13	0.011408078	0.048743606

## Data Availability

The sequencing data files were submitted to The Sequence Read Archive under Submission ID: SUB12168115, and BioProject ID: PRJNA894203. Upon availability, the data can be accessed through http://www.ncbi.nlm.nih.gov/bioproject/894203, accessed on 15 June 2022.

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
