# Peer review of "Gut Microbiome Remains Static in Functional Abdominal Pain Disorders Patients Compared to Controls: Potential for Diagnostic Tools"

_biotech, 2022, doi:10.3390/biotech11040050_

Round 1

Reviewer 1 Report

The manuscript is interesting. The proposal could have clinical interest. However, I have the following comments.

I. Major Comments:

1. It is necessary to include information about the volunteers studied, for example, age, weight, height, BMI, gender, etc.

2. They evaluated the diet of the subjects. It would be very good to increase the quality of the manuscript to include this information.

3. It is necessary to improve the discussion, specifically the points that allow the findings to be related to a possible clinical application (diagnosis).

II. Minor comment:

1. Improve the writing of the objective of the study.

Reviewer 2 Report

General comments:

This study by Abomoelak et al. is entitled "Gut microbiome remains static in functional abdominal pain disorders patients compared to controls: Potential diagnostic tools". It is nicely written but it is too descriptive and the use of the results as diagnostic tools is not described in details and therefore not convincing.

In addition the added value of the present study compared to their previous paper (Abomoelak et al. "The Gut Microbiome Alterations in Pediatric Patients with Functional Abdominal Pain Disorders" in Microorganisms, 2021. 9(11)) is not clearly stated. In that paper the authors apparently used the very same cohort to also study their microbiome comparing school term to vacation.

Specific comments:

-abstracts and keywords mention "PROMIS stress scores" as well as T-score but this is not presented in the main text;

-Figure 1 mention "N" for both the number of patients and the number of stool samples. It should be reported differently for the sake of clarity;

-As core results, at least a short description of the stool sequencing should be given in the material and methods part;

-Table 2: "Coefficient" is not defined in the legend. "N.not.0" is not defined in the legend. In addition, why mention both p-value and q-value if only p-value is considered for significance assessement?

-Line 160: a bracket is missing;

-Line 269: authors mentioned "probiotics" in the conclusion while this isn't mentioned anywhere in the main text. This should be extended if really relevant to the study.

Reviewer 3 Report

The manuscript entitled Gut microbiome remains static in functional abdominal pain 2 disorders patients compared to controls: Potential for diagnostic tools represents an adequate method for differentiating FAPDS vs Control. However, I have following suggestions/comments for the study

1. Low sample size with IBS issue with some of the samples.

2. Did the authors try to evaluate the expression for another markers from the stool, for an example-qPCR for selected target genes from fecal samples.

3. Please elaborate the findings on alpha diversity and beta diversity.

4. Please conclude the study as a take home massage.

Reviewer 4 Report

Dear Authors,

Please find below my comments on your manuscript. 

Background / Introduction

Excellent standard, with some comments. Please see point 2. below. 

Methods

Methods are appropriate and adequately explained. 

Results 

Clearly presented in adequate amount of detail. 

Suggestions for improvement

1. Given the significant differences in microbiome composition between school terms and summer vacation in healthy controls, the fact that the PROMIS stress questionnaire was only completed by the FAPD group is a missed opportunity. It would have been interesting to assess the impact of stress on the changes in microbiota experienced by healthy controls, and to compare the stress scores between HC and FAPD groups. I recommend that this be added to the manuscript as a limitation to the study. 

2. The PROMIS stress questionnaire is mentioned in line 21 but no further detail is provided. There is an opportunity here to explain your rationale behind the choice of the PROMIS stress measure as a self-reporting tool. Additionally, correlations between patient-reported outcomes and microbiome composition changes is an approach that adds a person-centred angle to your study. Please expand on these points as you have an opportunity to guide others who may wish to follow similar approaches in the future.

3. Additionally, in lines 245 to 251 you write that "the main butyrate-producing bacteria, Faecalibacterium, Roseburia, and Eubacteria, were not among the 55 bacterial species that showed changes." Then: "When we further studied the smaller groups with paired samples, only control samples showed changes in the two most abundant species, Bacteroides(increase) and Faecalibacterium (decrease) during the school term (Figure 3A)." 

If there is a decrease in Faecalibacterium, then the text in lines 245-246 is incorrect and confusing. Please review. 

I hope you find my comments valuable. 

Kindest regards,

The reviewer

Round 2

Reviewer 1 Report

The requested information is important for the scientific quality of the manuscript. Unfortunately the authors did not respond to my questions.

Author Response

The authors added the diet as a limitation of the study in the discussion in the revised manuscript. 

Reviewer 2 Report

authors addressed my comments in a satisfactory way

Author Response

Thanks so much.